# Cohort profile: Health trajectories of Immigrant Children (CRIAS)–a prospective cohort study in the metropolitan area of Lisbon, Portugal

Zélia Muggli [ID],[1] Thierry Mertens,[1] Regina Amado,[1] Ana Lúcia Teixeira [ID],[2] Dora Vaz,[3] Melanie Pires,[3] Helena Loureiro,[4] Inês Fronteira [ID],[1] Ana B Abecasis [ID],[1] António Carlos Silva,[5,6] Maria Rosário O Martins [ID][1]

For numbered affiliations see end of article.

**Correspondence to**
Professor Maria Rosário O Martins; mrfom@ihmt.unl.pt and Professor Maria Rosário O Martins, Global Health and Tropical Medicine (GHTM), Institute of Hygiene and Tropical Medicine (IHMT), NOVA University of Lisbon, Lisbon, Portugal; mrfom@ihmt.unl.pt

## ABSTRACT

**Purpose** The CRIAS (Health trajectories of Immigrant Children in Amadora) cohort study was created to explore whether children exposed to a migratory process experience different health risks over time, including physical health, cognitive, socioemotional and behavioural challenges and different healthcare utilisation patterns.

**Participants** The original CRIAS was set up to include 604 children born in 2015, of whom 50% were immigrants, and their parents. Recruitment of 420 children took place between June 2019 and March 2020 at age 4/5 years, with follow-up carried out at age 5/6 years, at age 6/7 years currently under way.

**Findings to date** Baseline data at age 4/5 years (2019–2020) suggested immigrant children to be more likely to belong to families with less income, compared with non-immigrant children. Being a first-generation immigrant child increased the odds of emotional and behavioural difficulties (adjusted OR 2.2; 95% CI: 1.06 to 4.76); more immigrant children required monitoring of items in the psychomotor development test (38.5% vs 28.3%). The prevalence of primary care utilisation was slightly higher among immigrant children (78.0% vs 73.8%), yet they received less health monitoring assessments for age 4 years. Utilisation of the hospital emergency department was higher among immigrants (53.2% vs 40.6%). Age 5 years follow-up (2020–2021) confirmed more immigrant children requiring monitoring of psychomotor development, compared with non-immigrant children (33.9% vs 21.6%). Economic inequalities exacerbated by post-COVID-19 pandemic confinement with parents of immigrant children 3.2 times more likely to have their household income decreased.

**Future plans** Further follow-up will take place at 8, 10, 12/13 and 15 years of age. Funds awarded by the National Science Foundation will allow 900 more children from four other Lisbon area municipalities to be included in the cohort (cohort-sequential design).

## STRENGTHS AND LIMITATIONS OF THIS STUDY

⇒ The CRIAS cohort is the first study of children in the Lisbon area, Portugal providing longitudinal data and insights about the little-known health trajectories of immigrant children in this country, with the potential to identify early interventions.
⇒ Strategic partnership between the university, National Health Service and a non-governmental organisation, generating a unique repository linking data from health centres, hospital and face-to-face questionnaires collected over time.
⇒ The study allowed the follow-up of vulnerable families during the COVID-19 pandemic in Amadora.
⇒ Absence of information on children who do not attend primary healthcare limits the representativeness of the study to those who attend public primary care.
⇒ Recruitment of children stopped 3 months before planned due to COVID-19 pandemic restrictions resulting in a smaller sample size. The pandemic context has also been a key challenge to the first follow-up.

## INTRODUCTION

For the purpose of the CRIAS cohort, an immigrant child was defined as a child residing in Portugal and born in a non-European Union (EU) country (first-generation immigrant) or having one or both parents born in a non-EU country; a non-immigrant child was born in Portugal to both parents born in Portugal.

In Portugal, 6.4% (662 095) of the population in 2020 was made up by foreign nationals. The majority (69%) are non-EU nationals and due to Portugal's colonial past, originating mostly from Brazil (28%) and Portuguese-speaking countries in Africa (14.4%), with an increasing number arriving from Asia.[1]

Health effects of migration processes are complex and the need to increase the knowledge base, especially for more vulnerable groups like children, has been highlighted at the national and international levels.[2–4] European studies report immigrant children often live in low-income and socially disadvantaged environments which can

adversely impact their health outcomes.[4–6] In Portugal, poor socioeconomic conditions among immigrants, compared with the non-immigrant population, have also been reported.[7–9] Comparison between studies and generalisation of research findings can be difficult because of the diversity in the definition and contexts of immigrant children across the EU. However, the general trend suggests that immigrant children present distinct health needs and more frequent health problems,[10 11] including being more at risk of overweight, obesity and some infectious diseases.[12 13] Inequalities in access and utilisation of healthcare services were observed with immigrant children having less probability of having a regular healthcare provider and using dental services but using more frequently hospital emergency departments compared with non-immigrant children.[14] Lower vaccination coverage[15–17] and emotional and behavioural difficulties[18–21] appear to be more frequent among immigrant children compared with non-immigrant children.

Childhood, especially the first 8 years, encompasses a period of rapid growth and development which plays a key role for health and well-being throughout the life course.[22] This period is highly influenced by the environment where the child grows and develops, and in particular, by socioeconomic factors.[23] Therefore, gaining evidence on children's health and development profiles, during the preschool period and following them in a longitudinal study, provides the possibility to formulate and implement early interventions to reduce inequalities.[24 25] These can help children not only to reach their full potential when starting school, but also to have a positive impact on their own future health, well-being and educational trajectories and also potentially impact their offspring through transgenerational effects.[26]

The first 3–5 years of life appear to be an opportunity window for ensuring adequate nutrition and physical exercise, for promoting parenting quality, child and parents' mental health, social–emotional competencies and language and communication skills which are linked to school readiness and better health later on.[24 27] Studies in the UK suggest that it is possible to mitigate poor outcomes with adequate family support services and interventions in schools.[24]

Several birth cohorts where data on immigration status or ethnicity have been collected have been established in Europe. Whereas some have compared health risks and outcomes between immigrant and non-immigrant children such as in Germany,[28] the UK,[29] the Netherlands,[30] France[31] and Spain,[6] others have not used the migration status for comparative analyses.[32] Many birth cohorts include only a small proportion of immigrant children, a scoping review reported an average of 10% (ranging from 0% to 60%) of migrant participation in birth cohorts in Europe.[32] In Portugal, for instance, only one birth cohort study, established in 2005, is conducted in the northern region of the country (Porto area) with a focus on the study of fetal and childhood determinants in the development of obesity and eventual metabolic changes. This

study also examined the relationship between migration and breast feeding and adverse pregnancy outcomes.[8 33] Like other European birth cohorts, it includes only a small proportion of immigrant children.[8]

The CRIAS cohort is the first longitudinal study in the metropolitan area of the capital city Lisbon that specifically focuses on gaining a better understanding of the health and development trajectories of immigrant and non-immigrant children, given their respective socioeconomic and cultural contexts. The aim of the CRIAS cohort is to explore whether children exposed to a migratory process present with different physical health outcomes, cognitive, socioemotional and behavioural challenges and with different healthcare utilisation patterns, over time, when compared with children born in Portugal and raised by parents also born in Portugal. The development of this cohort arises from a strategic and unique partnership between the university, the National Health Service and AJPAS (Associação de Intervenção Comunitária, Desenvolvimento Social e de Saúde)—a local non-governmental organisation (NGO) focusing on the needs of immigrant populations. This paper describes the characteristics of the cohort, the baseline cross-sectional study's and the first follow-up main findings.

## COHORT DESCRIPTION

### Setting

The CRIAS Study is conducted in the Amadora municipality, in the metropolitan area of Lisbon, Portugal. With 171 500 inhabitants in 2021,[34] it is the most densely populated municipality in the country. With a history of immigrant settlement, 13% of its population had a foreign nationality in 2020, making Amadora the second municipality in Portugal with the highest density of foreign residents—977/km$^2$.[1 35] It is currently served by 10 primary healthcare centres (nine up to December 2020), from now on referred as health centres, and 1 referral hospital—Hospital Professor Doutor Fernando Fonseca. The National Health Service in Portugal (SNS), based on the Beveridge model, is universal and free for children up to the age of 18 years. Hence, healthcare arrangements in the SNS are the same for all children regardless of their migration status. They include preventive measures such as vaccination and child health monitoring assessments carried out in health centres, as well as specialist and hospital care.

### Recruitment and participants

Recruitment was scheduled to take place in the nine health centres between June 2019 and June 2020. However, due to the COVID-19 pandemic, it was discontinued in March 2020. Children were recruited at age 4–5 years in order for the study to have as many children as possible born outside of Portugal and to be able to identify interventions in all children before school age (6 years). To be included in the study, children had to be born in 2015 and to have records of attending the health centre in the

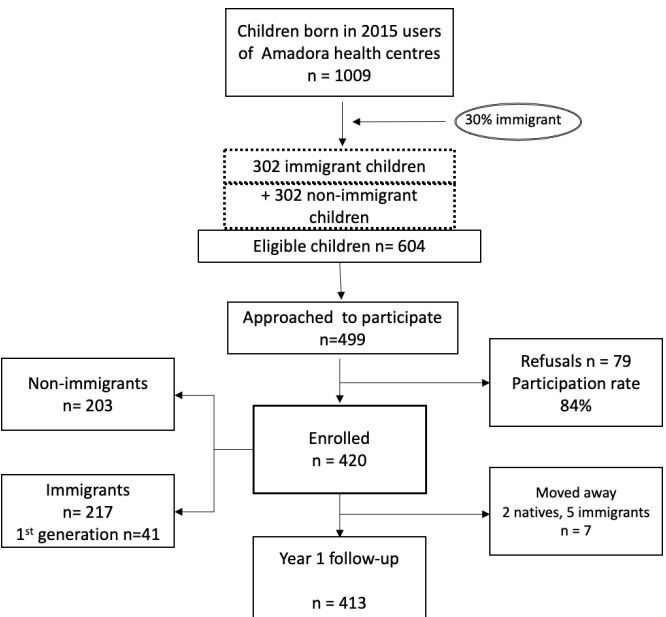

**Figure 1** Flow diagram of CRIAS cohort participants.

previous 2 years. There were 1009 children with these eligibility criteria in 2019. Based on a previous study,[36] we assumed that around 30% of users were immigrant children, that is, 302. In order to maximise comparisons over time between immigrant and non-immigrant children, we sought to have the same number of each, resulting in a total of 604 children eligible to participate, together with parents/caregivers.

Families were enrolled while in attendance at the health centre. Recruitment weeks were randomly distributed among the nine health centres, and the number of children recruited was proportional to the number registered in each centre. During recruitment, 499 parents/caregivers were approached; participation rate was 84%. From the 420 children enrolled, 217 were immigrants (51.6%) and 203 non-immigrants, 6 children were twins (4 immigrants, 2 non-immigrants).

At age 5/6 years follow-up, children's health centre records showed that seven children had moved to another municipality. Figure 1 illustrates cohort participation.

Starting in January 2022, families are being contacted by phone to arrange further follow-up assessments.

In order to facilitate enrolment and minimise losses to follow-up, several strategies were implemented:

▶ A pilot study (n=33) was conducted to verify acceptability of the questionnaires in terms of content and time by the families.
▶ Active engagement with health professionals. An official presentation of the study to all staff took place in a public venue, followed by further kick-off meetings in each health centre where an interlocutor was nominated to interact with the study team.
▶ An international team of six researchers from five different Portuguese-speaking countries, and proficient in six different languages, was trained to carry out recruitment and conduct initial interviews.

▶ All participants received details on the study objectives and direct contacts of the research principal investigator. Confidentiality issues and other questions raised were addressed in a culturally sensitive manner by the researchers. Interviews were conducted in total privacy in specially allocated rooms.
▶ Communication with parents is kept by phone each year and the contacts database is updated regularly with information from the health centres. Feedback (post/email) on screening outcomes is provided and when needed direction to further assessments is given. Face-to-face contact is preferred whenever possible.
▶ A local NGO AJPAS working with supporting immigrant communities is involved to facilitate participation.

## Data collection
The first wave of data collection at age 4/5 years was carried out in health centres between June 2019 and March 2020 by a team of six researchers using structured questionnaires. All interviewers received the same detailed information and training on the interview process.

Face-to-face interviews were held with parents/caregivers and as a first step, we collected family's socioeconomic and demographic characteristics migration history and child health information. The interviews were conducted mostly in Portuguese with other languages used when needed, for example, Creole, English or Asian languages. This was followed by a self-administered screening questionnaire—the Strengths and Difficulties Questionnaire (SDQ), available in validated translations and administered in the preferred language of the participant.

By June 2020, 85% of all COVID-19 infections in Portugal were concentrated in the metropolitan area of Lisbon, Amadora being one of the most affected municipalities.[37] Lockdown was declared in March 2020. To explore the socioeconomic dynamics of the cohort families during the COVID-19 pandemic, an intermediate data collection was undertaken in July 2020. Phone interviews were conducted applying a semistructured questionnaire exploring eventual changes in employment and household income, material deprivation and difficulties related to healthcare access.

The restriction measures adopted during the first waves of the COVID-19 pandemic have limited access to health centres and delayed collection of follow-up data. Nevertheless, baseline (children aged 4/5 years) and first follow-up (children aged 5/6 years) clinical data from electronic records for primary care and hospital emergency department visits were collected from November 2019 to October 2021.

## Instruments and variables
A schematic representation of instruments and variables used on data collection is shown in figure 2. Parents/caregivers were interviewed using a pilot-tested structured questionnaire to collect sociodemographic information

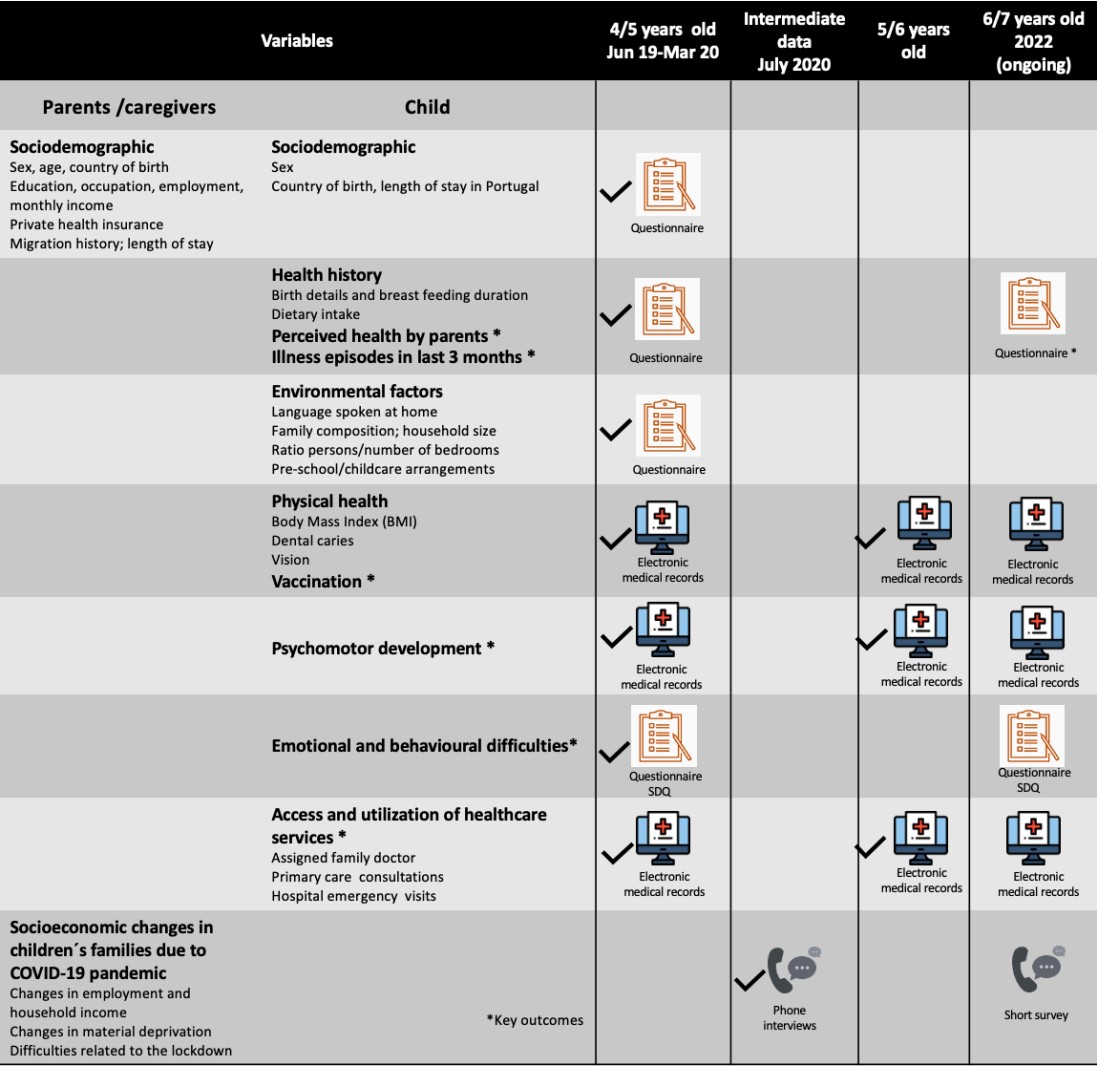

**Figure 2** Representation of data collection and main variables of the CRIAS cohort study.

on parents and children; child health history and environmental factors.

The SDQ is a brief questionnaire to assess child emotional and behavioural difficulties, self-administered to one parent or main caregiver, in the parents version for 4–17 years old, translated to Portuguese by Fleitlich et al[38] as well as in validated translations in other languages. It has been widely used and validated in research, including in multiethnic populations of children[39 40] and in several countries including Portugal.[41] It consists of five subscales, four measuring difficulties which can be grouped into two broad categories of behaviours: externalising (conduct problems+hyperactivity) and internalising (emotional+peer problems) behaviours. One subscale measures a strength—prosocial behaviour. A total score of difficulties can also be calculated to classify results as normal, borderline or at risk.

Physical health and development information for the ages of 4 and 5 years old was retrieved by two medical doctors who are part of the team from the medical records available on the SCLINICO primary healthcare information system. Measurements were performed during the

child health monitoring assessments by health professionals in line with the National Child and Youth Health Programme guidelines.[42] Psychomotor development was evaluated by the modified Mary Sheridan screening test, during the health monitoring assessments for ages 4 and 5 years.[42] This screening scale is used as a reference standard for several skills, distributed in four domains: (1) posture and global motor skills; (2) vision and fine motor skills; (3) hearing and language; (4) behaviour and social adaptation. Outcomes of the test were categorised in 'monitoring not required' if all items were fulfilled and in 'monitoring required' if one or more items were not achieved and required a review, usually carried out in 6 months' time. Additional variables from electronic medical records were related to access and utilisation of healthcare services; data on utilisation of the emergency department were provided by the hospital.

### Data linkage

Electronic medical records are managed in primary healthcare and in the hospital in two different information systems: SCLINICO and SORIAN, respectively.

We link three different datasets: data collected through interviews; primary healthcare data and hospital data. The information is obtained from the electronic medical records with the SNS number which is then returned to the project coordinator who matches this number with the identification (ID) code of the child. The key which assigns the SNS user number to the name of the child and to the ID code is password protected and kept by the coordinator of the study. The integrated cohort database available for analysis only includes the ID code.

## Patient and public involvement

The NGO AJPAS, founded by immigrants and located in Amadora municipality, and the members of the regional health authorities have been involved in the design, governance and general oversight of all phases of the research to date. Study participants have been encouraged to communicate to the research team by phone and email. Reports and presentations are frequently shared with key stakeholder groups. Members of the NGO AJPAS have been trained to also participate as interviewers in the survey on the socioeconomic impact of the COVID-19 pandemic providing economic, social and legal support whenever requested by the study participants.

## FINDINGS TO DATE

### Baseline characteristics of children and parents/caregivers

A large majority of the 417 parent/caregivers interviewed were women (88%), nearly all were the mothers. The main countries of origin of immigrant parents/caregivers were Cape Verde (n=60), Angola (n=28), Brazil (n=28) and Guinea-Bissau (n=22). The main reasons for immigration given by the mother were family reunification (28.9%), obtaining better education (27.6%), economic reasons (22.4%), with 3.3% having moved because of war; the median length of stay in years in Portugal was 9 years (min=0.1–max=37). Information on the main sociodemographic characteristics of the families is found in table 1.

Information collected on 217 (51.6%) immigrant children in the cohort showed that 41 children were born in a non-EU country. They originated mainly from the community of Portuguese-speaking countries: Brazil (13), Angola (8), Guinea-Bissau (6) and Cape Verde (4). The median length of stay in Portugal of these first-generation immigrant children was 18 months (min=1–max=48). Children from countries such as India, Nepal or Eritrea are also present in the study. Portuguese is spoken in 268 households (64%) while 17 other languages, ranging from Nepalese to Mandarin and Tigrinya, are spoken in the remaining households. After Portuguese, the most common language spoken is a combination of Creole and Portuguese (20%), spoken not only by immigrant families but also in 6.4% of households of non-immigrant children, suggesting a possible migration background of the grandparents. Table 2 shows other relevant characteristics of participating children.

## Findings on key outcomes and other variables at ages 4/5 and 5/6 years

At baseline (age 4/5 years), the perceived health of the child was considered to be very good or good by 80% of the parents. The median number of parent-reporting episodes of illness in the last 3 months was one and did not differ between immigrant and non-immigrant children. Most frequently reported complaints for immigrant and non-immigrant children, respectively, were related to cough and other symptoms of the respiratory tract (49.5% vs 66.5%), fever (18.8% vs 17.3 %), skin problems (6.4% vs 4%) and digestive complaints (7.9% vs 6.5%). Vaccination rates were above 90% for all children. In the Modified Mary Sheridan test to evaluate psychomotor development, more immigrant children were found to require monitoring of one or more items (38.5% vs 28.3%). In both groups, 25% of children required monitoring in items on the vision and fine motor skills domain. The above information is shown in online supplemental table 1. The findings on emotional and behavioural difficulties suggest that a low family income (adjusted OR (aOR) 4.5; 95% CI: 1.43 to 13.95), low parental education level (aOR 2.5; 95% CI: 1.11 to 5.16) and being a first-generation immigrant child (aOR 2.2; 95% CI: 1.06 to 4.76) may increase significantly the odds of developing emotional and behavioural difficulties; these results are shown in online supplemental table 1B.

The main variables collected on health services utilisation are summarised in table 3. Over a quarter (26%) of immigrant children did not have a regular allocated family doctor Non-immigrant children used less primary care (73.8% vs 78%); the hospital emergency department was more used by immigrant children (53.2% vs 40.6%).

Other findings at age 4/5 years (online supplemental table 2) included dental caries observed in 23% of the children with a similar number having vision acuity or eyes alignment difficulties, with no differences among groups. The recommended intake of fruits and vegetables is not achieved by most children, particularly immigrant children. Overweight was found in 25.2% of the children (22.2% in immigrant vs 28.2% in non-immigrant children), 5.4% of children were obese and from a total of 8.3% underweight children, most were immigrant.

The main results from the additional module (July 2020) on the potential socioeconomic effects of COVID-19 pandemic on families participating in the CRIAS cohort study are shown in online supplemental table 3. Immigrant parents were more likely to be unemployed due to the COVID-19 pandemic (aOR 3.54, 95% CI 1.72 to 7.30) and more likely to have their household income decreased (aOR 3.21, 95% CI 1.80 to 5.75).

At age 5 years follow-up, during the first year of the COVID-19 pandemic, about two-thirds of all children did not receive routine health monitoring assessments for age 5 years, mostly due to limited access to the health centres as a result of the pandemic restrictions. Immigrant children continued to require greater attention on their psychomotor development with 33.9% vs 21.6%

**Table 1** Baseline characteristics of the parents/caregivers of children in the CRIAS cohort

| Parents/caregivers n=417 | Immigrant children n (%) | Non-immigrant children n (%) | Total n (%) | P value |
|---|---|---|---|---|
| Sex | | | | 0.440* |
| Female | 187 (86.6) | 179 (89.1) | 366 (87.8) | |
| Age, n=417 | | | | 0.213** |
| Median (min–max; IQR) | 34 (20–75; 10) | 35 (18–68; 10) | 35 (18–75; 10) | |
| Relationship with child, n=417 | | | | 0.262* |
| Mother | 182 (84.3) | 177 (88.1) | 359 (86.1) | |
| Others | 34 (15.7) | 24 (11.9) | 58 (13.9) | |
| Educational level*, n=416 | | | | 0.115* |
| Lower education | 40 (18.6) | 27 (13.4) | 67 (16.1) | |
| 9 years completed | 41 (19.1) | 45 (22.4) | 86 (20.7) | |
| Secondary education | 91 (42.3) | 73 (36.3) | 164 (39.4) | |
| University degree | 43 (20.0) | 56 (27.9) | 99 (23.8) | |
| Occupation†, n=414 | | | | <0.001* |
| High skilled | 34 (16.0) | 69 (34.3) | 103 (24.9) | |
| Medium skilled | 99 (46.5) | 102 (50.7) | 201 (48.6) | |
| Low skilled | 75 (35.2) | 20 (10.0) | 95 (22.9) | |
| Non-defined | 5 (2.3) | 10 (5.0) | 15 (3.6) | |
| Employment status, n=417 | | | | 0.009* |
| Employed with a contract | 135 (62.5) | 157 (78.1) | 292 (70.0) | |
| Employed without a contract | 20 (9.3) | 5 (2.5) | 25 (6.0) | |
| Unemployed with benefits | 13 (6.0) | 10 (5.0) | 23 (5.5) | |
| Unemployed without benefits | 18 (8.3) | 11 (5.5) | 29 (7.0) | |
| Self-employed | 16 (7.4) | 9 (4.5) | 25 (6.0) | |
| Other‡ | 14 (6.5) | 9 (4.5) | 23 (5.5) | |
| Household monthly income, n=395 | | | | <0.001* |
| <€500 | 39 (19.1) | 12 (6.3) | 51 (12.9) | |
| >€500–€750 | 66 (32.4) | 44 (23.0) | 110 (27.8) | |
| >€750–€1000 | 38 (18.6) | 34 (17.8) | 72 (18.2) | |
| >€1000–€1500 | 36 (17.6) | 43 (22.5) | 79 (20.0) | |
| >€1500–€2000 | 16 (7.8) | 24 (12.6) | 40 (10.1) | |
| >€2000 | 9 (4.4) | 34 (17.8) | 43 (10.9) | |

Significance level 5%. *Pearson X$^2$ statistical test; **Mann-Whitney U statistical test.
*Based on the International Standard Classification of Education.[47]
†Classified as per the Portuguese Classification of Professions and summarised in four skill levels according to the International Standard Classification of Occupations.[48]
‡Students, stay-at-home parents, retired.

non-immigrant children having test items with monitoring required.

Emergency department use dropped significantly (28.9% for immigrant children vs 26.7%). Table 3 compares healthcare utilisation at ages 4 and 5 years.

The findings on emotional and behavioural difficulties have been published[43]; general findings from the first wave of data have been published as an abstract in the European Journal of Public Health.[44] Results from the survey on the socioeconomic impact of COVID-19 on immigrant and non-immigrants families were awarded with the Human Rights Gold Medal Prize given in 2020 by the National Assembly of the Republic of Portugal; main findings have been submitted to a scientific journal and are under review.

## FUTURE PLANS

We are preparing to resume face-to-face contacts with families and are already contacting the participants by

**Table 2** Main characteristics of children in the CRIAS cohort at baseline age 4/5 years

| Characteristics of the children | Immigrant n (%) | Non-immigrant n (%) | Total n (%) | P value |
|---|---|---|---|---|
| | 217 (51.7) | 203 (48.3) | 420 (100) | |
| Sex n=420 | | | | 0.689* |
| Female | 109 (50.2) | 98 (48.3) | 207 (49.3) | |
| Gestational age, n=413 | | | | 0.463* |
| <37 weeks—preterm | 15 (7.0) | 18 (9.0) | 33 (8.0) | |
| >37 weeks | 198 (93) | 182 (91) | 380 (92) | |
| Birth weight, n=385 | | | | 0.531* |
| <2500 g—low birth weight | 19 (9.9) | 16 (8.1) | 35 (9.0) | |
| >2500 g | 172 (90.1) | 181 (91.9) | 353 (91.0) | |
| Breast feeding, n=419 | 203 (93.1) | 179 (89.1) | 382 (91.2) | 0.152* |
| Total duration breast feeding, n=375, median months (min–max; IQR) | 12 (0–53;18) | 6 (0–48;14) | 10 (0–53;15) | <0.001** |
| Family structure, n=419 | | | | 0.084* |
| Both parents | 99 (45.8) | 117 (57.6) | 216 (51.6) | |
| Both parents and others | 30 (13.9) | 27 (13.3) | 57 (13.6) | |
| Single-parent families | 42 (19.4) | 28 (13.8) | 70 (16.7) | |
| One parent and others/others | 45 (20.9) | 31 (15.3) | 76 (18.1) | |
| Large households (≥5 people), n=420 | 79 (36.2) | 57 (28.2) | 136 (32.4) | 0.079* |
| Ratio of people in household/number of bedrooms, n=420 mean (95% CI) | 2.00 (1.89 to 2.11) | 1.73 (1.65 to 1.81) | 1.87 (1.80 to 1.94) | <0.001*** |
| Childcare arrangements, n=417 | | | | 0.329* |
| State preschool | 83 (38.4) | 81 (39.9) | 164 (39.1) | |
| Private preschool | 94 (43.5) | 98 (48.3) | 192 (45.8) | |
| Stays home w/ mother | 15 (6.9) | 8 (3.9) | 23 (5.5) | |
| Other | 24 (11.1) | 16 (7.9) | 40 (9.5) | |
| Assigned family doctor, n=420 | 161 (73.9) | 179 (88.6) | 340 (81.0) | <0.001* |
| Private health insurance, n=417 | 63 (29.3) | 104 (51.5) | 104 (51.5) | |

Significance level 5%. *Pearson $X^2$ statistical test; **Mann-Whitney U statistical test; ***t-test.

phone. Selected socioeconomic information will be updated. Considering the recent rise in mental health difficulties in children, related with the COVID-19 pandemic, the follow-up SDQ assessment might reveal new developments. An additional module to study asthma and allergies in childhood will be implemented at age 6/7 years, using International Study of Asthma and Allergies in Childhood methodology.[45] Information on the experience of accessing and using health services by the immigrant children in the study and their families will be complemented by a qualitative study. Further follow-ups will be carried out at the key ages of 8, 10, 12/13 and 15/18 years of the National Child and Youth Health Programme[42] if ongoing financing.

The conduct of this study and its societal implications led us to extend the study to another four municipalities in the Lisbon region with the collection of data on further 900 children (450 immigrants) through a sequential-cohort design, likely to include more immigrants from non-lusophone countries. Funded by the National Science Foundation, in partnership with 2 local NGOs (AJPAS and Doctors of the World) and 15 health centres, the extension of the cohort study will start in February 2022.

We will continue to disseminate our results in conferences, scientific papers and meetings with local NGOs and policymakers at the regional level. A book is in preparation for the Migrations Observatory in Portugal.

Although Portugal provides free healthcare for all children including undocumented migrants and repeatedly scores high in migrant integration policies,[46] the Migration Policies Index gives a less favourable score for healthcare. Therefore, we will continue to work to translate our findings into policies and services change to improve access and quality of healthcare provision, and contribute to better lives of all children.

## STRENGTHS AND LIMITATIONS OF THIS STUDY
Our study presents several strengths. First and foremost, CRIAS is the first cohort study in the metropolitan area of the capital city Lisbon, Portugal, created to

**Table 3** Utilisation of healthcare services by children in the CRIAS cohort at age 4 and 5 years

| | 1st wave of data collection (age 4) n=420 | | | | 2nd wave of data collection (age 5) n=420 | | | |
|---|---|---|---|---|---|---|---|---|
| | Immigrant n (%) | Non-immigrant n (%) | Total n (%) | P value | Immigrant n (%) | Non-immigrant n (%) | Total n (%) | P value |
| **Primary care** | **At least one consultation in 2019*** | | | 0.312* | **At least one consultation in 2020†** | | | 0.018* |
| | 170 (78.0) | 149 (73.8) | 319 (76.0) | | 153 (70.2) | 162 (80.2) | 315 (75.0) | |
| **Most frequent diagnosis in 2019‡** | | | | | **Most frequent diagnosis in 2020‡** | | | |
| **Respiratory infections** | 52 (24.0) | 58 (28.6) | 110 (26.2) | 0.283* | 20 (9.2) | 33 (16.3) | 53 (12.6) | 0.027* |
| **Skin** | | | | | | | | |
| Parasitic and fungal infections | 8 (3.7) | 3 (1.5) | 11 (2.6) | 0.157** | 7 (3.2) | 3 (1.5) | 10 (2.4) | 0.341** |
| Atopic dermatitis | 31 (14.3) | 14 (6.9) | 45 (10.7) | 0.014* | 17 (7.8) | 12 (5.9) | 29 (6.9) | 0.453 |
| **Digestive** | | | | | | | | |
| Gastroenteritis | 11 (5.1) | 11 (5.4) | 22 (5.2) | 0.872* | 5 (2.3) | 5 (2.5) | 10 (2.4) | 0.903* |
| Others | 23 (10.6) | 13 (6.4) | 36 (8.6) | 0.125* | 16 (7.3) | 16 (5.9) | 32 (7.6) | 0.822* |
| | **Health monitoring assessment at age 4** | | | | **Health monitoring assessment at age 5** | | | |
| | 161§ (73.9) | 160 (79.2) | 321 (76.4) | 0.196* | 72¶ (33.2) | 64 (31.5) | 136 (32.4) | 0.769* |
| **Hospital** | **At least one emergency department visit in year 2019** | | | | **At least one emergency department visit in year 2020** | | | |
| | 116 (53.2) | 82 (40.6) | 198 (47.1) | 0.010* | 63 (28.9) | 54 (26.7) | 117 (27.9) | 0.621* |

Significance level 5%. *Pearson $X^2$ square statistical test; **Fisher's exact test.
*Total number of consultations in 2019, n=735.
†Total number of consultations in 2020, n=606.
‡Based on the International Classification of Primary Care (ICPC-2). [49]
§36.6% of the first-generation immigrant children did not receive the health monitoring assessment for those aged 4 years old.
¶75.6% of the first-generation immigrant children did not receive the health monitoring assessment for those aged 5 years old.

specifically address the role of immigration as a physical and emotional health determinant, by comparing outcomes in immigrant and non-immigrant children over time. In contrast to many cohort studies initiated in the EU, which report results regarding immigration's potential impacts while including on average 10% of children with a migration background, this cohort study includes about 50%. This proportion of immigrant children in the sample increases the power of comparisons between immigrant and non-immigrant children over time compared with other studies. Due to the colonial past of Portugal, these immigrant populations are mainly from Brazil and Portuguese-speaking African countries who might have diverse migration experiences, biological and cultural factors and health utilisation behaviours than the populations included in other cohort studies in Europe.

The partnership with a local NGO is critical in the recruitment and follow-up phases and in providing direct support to immigrant families and their children during COVID-19 times. Timely presentation of the results to primary healthcare professionals potentiates the identification of early interventions. The start of the project just before the COVID-19 pandemic made it possible to follow and, whenever possible, support (via the NGO partnership) more vulnerable families during the pandemic crisis.

One limitation may appear during the next follow-up steps because of the early interruption of recruitment of eligible immigrant children (expected n=302) due to lockdown and social distancing restrictions required by the COVID-19 pandemic, resulting in a smaller sample size. Difficulties on re-establishing face-to-face contact with families and on providing the child health monitoring assessments by health professionals, often under constrained time and resources in the pandemic context, have delayed follow-up data collection. Another limitation is the absence of information on children who do not attend public primary healthcare centres which limits the representativeness of the study to those who attend public primary care.

**COLLABORATION**

Initial data analysis and publications will be generated by investigators on CRIAS cohort research team. Study data are not currently freely available. However, deidentified data are available upon reasonable request from the coordinator of the study (MROM-ORCID ID: 0000-0002-7941-0285). The research team welcomes collaboration with other researchers.

**Author affiliations**
¹Global Health and Tropical Medicine (GHTM), Institute of Hygiene and Tropical Medicine (IHMT), NOVA University of Lisbon, Lisbon, Portugal
²Interdisciplinary Centre of Social Sciences (CICS.NOVA), Faculty of Social Sciences and Humanities (NOVA FCSH), NOVA University of Lisbon, Lisbon, Portugal
³Amadora Primary Care Health Centres Group, Regional Health Administration of Lisbon and Tagus Valley, Ministry of Health, Lisbon, Portugal
⁴Paediatrics Department, Hospital Professor Doutor Fernando da Fonseca, Amadora, Portugal
⁵Public Health Department, Regional Health Administration of Lisbon and Tagus Valley, Ministry of Health, Lisbon, Portugal
⁶AJPAS-Associação de Intervenção Comunitária, Desenvolvimento Social e de Saúde, Amadora, Portugal

**Correction notice** This article has been corrected since it first published. Author name 'Ana Abecassis' is changed to 'Ana B Abecassis'.

**Acknowledgements** We would like to thank all participating families and health centre's staff. The authors acknowledge all the support received from the Executive Direction and Clinical Advisory Board of Amadora group of health centres and Hospital Professor Doutor Fernando Fonseca and the NGO AJPAS.

**Contributors** Conceptualisation—MROM, ZM, TM, ALT, DV, ACS, IF and AA. Methodology—MROM, ZM, TM and IF. Validation—MROM, ZM, RA, MP and HL. Formal analysis—MROM, ZM and RA. Data analysis—MROM, ZM and RA. Writing (original draft) preparation—ZM and MROM. Writing (review and editing)—MROM, ZM, TM, IF, AA, DV and ACS. Data collection—ZM, RA, MROM, MP and HL. Supervision—MROM. Project administration—MROM. Funding acquisition—MROM. Guarantor–MROM. All authors have read and agreed to the published version of the manuscript.

**Funding** This research was financed by the Asylum, Integration and Migration Fund (ref.PT/2018/FAMI/350) under the Multianual Financial Framework 2014/20, by the Portuguese Foundation for Science and Technology (FCT) (ref. RESEARCH4COVID-19-065) and Global Health and Tropical Medicine (GHTM), Institute of Hygiene and Tropical Medicine (IHMT), NOVA University of Lisbon, Portugal (ref.UID/04413/2020). The extension of the cohort study is financed by the Portuguese Foundation for Science and Technology (FCT) (ref.PTDC/SAU-SER/4664/2020).

**Competing interests** None declared.

**Patient and public involvement** Patients and/or the public were involved in the design, or conduct, or reporting, or dissemination plans of this research. Refer to the Methods section for further details.

**Patient consent for publication** Not required.

**Ethics approval** This study involves human participants and was approved by the Health Ethics Committee of the Regional Health Administration of Lisbon and Tagus Valley, Portugal (001/CES/INV/2019), including an additional approval for the COVID-19 intermediate study (9-2020/CES/2020). A written information and consent form to participate in the study was signed by the parents, which included permission to assess data from the child's health centre and hospital medical records.

**Provenance and peer review** Not commissioned; externally peer reviewed.

**Data availability statement** Data are available upon reasonable request. Deidentified participant data are available upon reasonable request from the coordinator of the study (MROM-ORCID ID: 0000-0002-7941-0285).

**ORCID iDs**
Zélia Muggli http://orcid.org/0000-0001-8557-9215
Ana Lúcia Teixeira http://orcid.org/0000-0002-8086-2254
Inês Fronteira http://orcid.org/0000-0003-1406-4585
Ana B Abecasis http://orcid.org/0000-0002-3903-5265
Maria Rosário O Martins http://orcid.org/0000-0002-7941-0285

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
