## [Reviewer comments · BMJ Open]

ARTICLE DETAILS

TITLE (PROVISIONAL)	COHORT PROFILE: HEALTH TRAJECTORIES OF IMMIGRANT CHILDREN – CRIAS, A PROSPECTIVE COHORT STUDY IN THE METROPOLITAN AREA OF LISBON, PORTUGAL
AUTHORS	Muggli, Zélia; Mertens, Thierry; Amado, Regina; Teixeira, Ana Lúcia; Vaz, Dora; Pires, Melanie; Loureiro, Helena; Fronteira, Inês; Abecassis, Ana; Silva, António; Martins, Maria Rosário

VERSION 1 – REVIEW

REVIEWER	Shuaijun Guo University of Melbourne School of Population and Global Health
REVIEW RETURNED	01-Apr-2022

GENERAL COMMENTS	Thank you for this opportunity to review this paper. It is a very interesting paper. After reading the full draft, I have two major concerns: one is the justification of doing this cohort. Particularly, why is it necessary to conduct this cohort, given a number of birth cohort studies have already collected data on migration and child health outcomes? The authors should give more details about the justification. For example, what is the unique strength of this cohort study that other birth cohort studies cannot achieve? The other concern is the sampling approach. The targeted sample was children who used health care services in the last two years. What is the percentage of this population in the total population of this metropolitan? Any census data that can support the representativeness? Also, the 1:1 pair approach is not clear and may bring about selection bias for the native children sample. From this side, I think the targeted sample is not all children, but those who were patients in the primary health care centres. Other minor comments are as follows: Abstract Line 4: Please spell the full name of CRIAS for the first time to use. Line 17-22: It is confusing for me about the first two sentences. Which year is the baseline recruitment (2015 or 2019)? If 2019 is the baseline data collection time, please mention it clearly. Also, the definition of immigrant and native children should be briefly introduced. Line 25-26: It is not clear of the meaning of “socioeconomic disadvantage”. Also please make the baseline data collection time consistent. It was age 4 here, but it was age 4 and 5 years previously. Line 27-28: It should be “The prevalence of primary care utilisation was higher among native children” Line 32: It is not clear for this sentence “More immigrant children had psychomotor development test items to monitor.” What do the authors mean for “test items to monitor”?
--

	Line 41-42: If “emergency room use” is same as “hospital emergency department”, please make it consistent. Could the authors just report the exact prevalence of emergency room use in the bracket, rather presenting with -45% and -32%? Line 48: The definition of “immigrant parents” should be also briefly introduced in Line 17-22. Strengths and limitations of this study Please use a full sentence to report each bullet point. Line 39-45: I do not think earlier recruitment is a limitation. What is the impact of the COVID-19 pandemic on the recruitment and follow-up engagement? Low response rate? Not representative samples? Line 48-51: The last point is not clear. If the authors mean the representativeness of the samples, please specify this. Introduction Page 4 Line 8-9: It would be much clear to list the total number of immigrants for the 6.4% in 2020, then readers will get an overall picture of the context. Page 4 Line 11: Please spell out EU for the first time to use. Page 4 Line 22-25: The first two sentences should be placed in a specific context. For example, was the evidence from previous studies conducted in Portugal? To my knowledge, immigrant children, particularly the 1st generation, are always from high socioeconomic status families in the USA and Australia, so they have better outcomes than native children (i.e., healthy migrant effect). Page 4 Line 40-48: I agree with the necessity of conducting research a a more regional level in a single country. But more justification is needed for the necessity of CRIAS. Currently, most birth cohort studies collected children’s and parents’ migration history data. So what is the strength of the CRIAS cohort that other birth cohorts cannot achieve? I think this is the most important part of the justification/rationale for this study. Maybe this reference is helpful (Grosser A, Razum O, Vrijkotte TG, Hinz IM, Spallek J. Inclusion of migrants and ethnic minorities in European birth cohort studies—a scoping review. The European Journal of Public Health. 2016 Dec 1;26(6):984-91.) Page 5 Line 10: pleas spell out CRIAS for the first time to use Page 5 Line 38: please spell out AJPAS for the first time to use Page 5 Line 42-56: I think this paragraph belongs to the Methods section about the participants. When I finish reading the setting section, I do not think “Children born in 2015, residing in the Metropolitan Lisbon Region -Amador, are followed up ...” is an accurate description, particularly for the word “followed up”, which means “children born in 2015 was recruited and baseline data were collected as well.” The authors mentioned later for the inclusion criteria “children had to be born in 2015” but the true recruitment time was 2019-2020. Page 5 Line 51-52: “Measurement were scheduled at the ages of 4,5,6/7 ...” should be more accurate. If a child was born in early 2015, and the recruitment occurred in Feb 2020, then the child age is 5 at baseline. So better to use age range for each measurement timing, such as 4/5, 5/6, 6/7 ... Cohort description Page 6 Line 19-20: The authors mentioned the Amador municipality is served by 10 primary health care centres. Do these 10 centres all
--	--

	public ones? Are there any private health care centres? Readers are not familiar with the health care system in Portugal. It would be great to briefly introduce whether it is universal health care or not. Page 6 Line 31: Did the recruitment discontinue in March 2020? If so, please make it clearer. Page 6 Line 34-42: What is the reason of selecting children who had to be born in 2015? The authors should justify why focusing on children aged 4-5 years at baseline. Also, the inclusion criteria based on the last two years' records of attending health care is another concern. That means, the recruiting samples are those who had used health services before. These children are not representative at all. Later on, the authors mentioned using 1:1 pair method for immigrant and native children. How did the authors pair this for native children? Based on a random selection from the same health care centre? Or from the same type of health service use? Or based on gender? Instruments and measures Page 9 Line 9-10: Please add response options in brackets for sex, age, country of birth and other variables, such as sex (male/female), age (continuous), ... Page 10 line 58-59: It would be better to describe the other group as "at risk" rather than "to monitor". Page 11 Line 3-17: I agree with the collection of these indicators about the access and utilisation of healthcare services. Will the authors also consider collecting data from the patients' perspective, such as health literacy, quality of health services? These factors are also very important to child health. Data linkage Please spell out the abbreviations for the first time to use, including SCLINICO, SORIAN, SNS, ID Table 1 It looks like native children are more likely to come from high socioeconomic status families (university degree: 27.9% versus 20.0%; high skilled: 34.3% vs 16.0%; >\$2000: 17.8% vs 4.4%). Is the difference true between immigrant children and native children? Or due to the selection bias (which I mentioned before for the 1:1 pair issue)? This should be discussed later. Table 2 What is the definition of "health coverage beyond national health service"? This term was not mentioned in the Methods section. Future plans Page 20 line 44-45: The context of "free health care for all children" should be introduced earlier before. Figure 1 Please check the accuracy of the first line entitled "children born in 2015 attending Amadora health care centres." Did the targeted children attend Amadora healthcare centres in 2015? If so, the baseline data collection time should be 2015. Supplementary table 1 What is the p-value for the psychomotor development? "normal" versus "at risk" would be better than "all items achieved" and "to monitor" as response options of psychomotor development.
--	--

REVIEWER	Mohammad Alasagheirin University of Wisconsin – Eau Claire, College of Nursing and Health Sciences
REVIEW RETURNED	10-May-2022

GENERAL COMMENTS	I think this manuscript addresses a very interesting and important question. The aim of the study as outlined on page 6 line 7 suggested a correlation between the exposures (which were not defined and measured) and the health outcomes of children. however, the result was mainly descriptive of the first wave of the cohort study. The discussion section needs more elaboration on the importance of the findings in relation to the children's health and as compared to other studies and research articles.
--

VERSION 1 – AUTHOR RESPONSE

REVIEWER 1

Thank you for this opportunity to review this paper. It is a very interesting paper.

Before providing a point -by-point response to comments , the authors would like to thank the reviewer for the time and consideration in reading and commenting this paper.

After reading the full draft, I have two major concerns: one is the justification of doing this cohort. Particularly, why is it necessary to conduct this cohort, given a number of birth cohort studies have already collected data on migration and child health outcomes? The authors should give more details about the justification. For example, what is the unique strength of this cohort study that other birth cohort studies cannot achieve?

Thank you for giving us the opportunity to further clarify these points. We have given more details on the justification to conduct this cohort in the introduction pg. 5 lines 13-20 and in the strengths and limitations section pg. 16-17 lines 24-4.

CRIAS is the first cohort study in Metropolitan Area of the capital city Lisbon, Portugal, and the first in the country, designed to specifically address the role of immigration as a physical and emotional health determinant, by comparing outcomes in immigrant and non-immigrant children over time. In contrast to many cohort studies in the EU (European Union) which have a small participation of immigrants, CRIAS includes about 50% of children with an immigration background . Due to the colonial past of Portugal, these immigrant populations are mainly from Brazil and Portuguese speaking African countries who might have diverse migration experiences, biological and cultural factors and health utilization behaviours than the populations included in other cohort studies in Europe. We believe that this set of characteristics makes this cohort unique.

The other concern is the sampling approach. The targeted sample was children who used health care services in the last two years.

This means that children born in 2015 and not using these services in the last two years before recruitment are excluded from the study. This is the result of our decision to recruit children at the health centres and we have highlighted it as a limitation (pg 16, lines 17-19).

What is the percentage of this population in the total population of this metropolitan? Any census data that can support the representativeness?

Based on Pordata, 1869 children were born in 2015 in Amadora; <https://www.pordata.pt/Municipios/Nados+vivos+de+m%c3%a3es+residentes+em+Portugal+total+e+por+sexo-103>. Based on our data, 54% (n=1009) used healthcare centres. Reasons for not using these services may include moving to live in another Municipality or attending the private health sector. We know that the number of families with 3-5 persons leaving in Amadora is decreasing over time: for example based on census data in 2011 there were 11% less families living in Amadora (with 3-5 persons) than in 2001. No data is available for 2021 census but the pattern is probably the same; this suggests that some of the children that are not using the services possibly moved to live in other municipalities (Pordata:<https://www.pordata.pt/Municipios/Fam%c3%adlias+cl%c3%a1ssicas+segundo+os+Censos+total+e+por+n%c3%bamero+de+indiv%c3%adduos-33>).

Data from 2017, suggests that at least 25% of the population had a second (or more) layer of health insurance coverage through health subsystems (for specific sectors or occupations) and voluntary private health insurance (Simões J, Augusto GF, Fronteira I, Hernández-Quevedo C. Portugal: Health system review. Health Systems in Transition, 2017; 19(2):1–184.)

We can suppose that nonusers are those who attend the private sector and those who moved to another municipality; these two groups have probably different characteristics.

We agree that there will be a bias related to this selection and describe it as a limitation page 16 line 17-19

Also, the 1:1 pair approach is not clear and may bring about selection bias for the non-immigrant children sample.

Thank you very much for this comment; it was not an easy choice to select the approach because we do not have data on the real percentage of non-immigrant and immigrant children attending health care centres. However a study conducted in 2018 (ref. 36 in text) suggested a figure around 30% of immigrant children. We could have chosen 2 non-immigrants for 1 immigrant (to be more close to the “real percentage”) but the expectation was to have greater loss to follow-up in immigrants (that are more frequently on the move) than in non-immigrants, and if so, 3 or 4 years later these percentages we could have a lower representation of immigrants. Actually, this is what we are observing now in CRIAS with 2.5% of immigrant children having moved away.

If the sample of non-immigrant children born in 2015 in Amadora and using health centers is representative, then the bias will be minimal. However, it is difficult to verify this assumption as there is no data for children’s health at the Municipality level, namely disaggregated by non-immigrants and immigrants.

From this side, I think the targeted sample is not all children, but those who were patients in the primary health care centres.

We agree with your comment , the target sample are children born in 2015 who were users in primary healthcare centres

Other minor comments are as follows:

ABSTRACT

Line 4: Please spell the full name of CRIAS for the first time to use. CRIAS is the acronym based on the Portuguese words for Crianças Imigrantes da Amadora e percursos de Saúde = Health trajectories of Immigrant Children in Amadora. It was spelt on the first line of abstract.

Line 17-22: It is confusing for me about the first two sentences. Which year is the baseline recruitment (2015 or 2019)? If 2019 is the baseline data collection time, please mention it clearly. Also, the definition of immigrant and non-immigrant children should be briefly introduced.

Recruitment took place from June 2019 to March 2020 when baseline data collection took place as well. We believe this was now better clarified on lines 8 and 10 in the abstract.

Due to the length of our definition of immigrant and non-immigrant children (we opted to change native to non-immigrant) and the word limite on the abstract , we included the definition in the first paragraph of the introduction section .” For the purpose of this study, an immigrant child was defined as a child residing in Portugal and born in a non-European Union (EU) country (1st generation immigrant) or having one or both parents born in a non-EU country; a non-immigrant child was born in Portugal to both parents born in Portugal.”

Line 25-26: It is not clear of the meaning of “socioeconomic disadvantage”. Also please make the baseline data collection time consistent. It was age 4 here, but it was age 4 and 5 years previously.

Socioeconomic disadvantage means in this context that immigrant children were more likely to belong to families with less income, more unemployment and less skilled occupations; again to use few words we changed that line to “Baseline data at age 4/5 showed immigrant children to be more likely to belong to families with less income” line 10

Line 27-28: It should be “The prevalence of primary care utilisation was higher among non-immigrant children” – We introduced this sentence and corrected values of primary care utilization line 14-15

Line 32: It is not clear for this sentence “”More immigrant children had psychomotor development test items to monitor.” What do the authors mean for “test items to monitor”?

We mean children who did not fulfil all items in the development test and will require to be reviewed to verify if those items are achieved before assessment at 5 years of age. This review is usually

carried out 6 months later depending on the context of the child We have explained this better explained in the methods section pg.9 , lines 4-6.

If “emergency room use” is same as “hospital emergency department”, please make it consistent. Could the authors just report the exact prevalence of emergency room use in the bracket, rather presenting with -45% and -32%? We have removed this results from abstract.

Line 48: The definition of “immigrant parents” should be also briefly introduced in Line 17-22. What we mean here is better written as “the parents of immigrant children”. Line 20 reads :” parents of immigrant children were 3.5 times more likely.....”

STRENGTHS AND LIMITATIONS OF THIS STUDY Please use a full sentence to report each bullet point.

We have incorporated this suggestion.

Line 39-45: I do not think earlier recruitment is a limitation. What is the impact of the COVID-19 pandemic on the recruitment and follow-up engagement? Low response rate? Not representative samples?

The COVID-19 pandemic resulted in earlier conclusion of recruitment and a smaller sample size. Concerning the follow-up, the very limited access to the health centres was an obstacle to data collection in most part of the years 2020 and 2021 . This is now clarified on the 5th bullet point

Line 48-51: The last point is not clear. If the authors mean the representativeness of the samples, please specify this. Representativeness is now addressed on bullet point 4

Introduction

Page 4 Line 8-9: It would be much clear to list the total number of immigrants for the 6.4% in 2020, then readers will get an overall picture of the context.

This number is 662,095. This is on pg.3 line 7 Page 4 Line 11: Please spell out EU for the first time to use.

This is now on page 3 line 4: non-European Union (EU) country

Page 4 Line 22-25: The first two sentences should be placed in a specific context. For example, was the evidence from previous studies conducted in Portugal? To my knowledge, immigrant children, particularly the 1st generation, are always from high socioeconomic status families in the USA and Australia, so they have better outcomes than non-immigrant children (i.e., healthy migrant effect).

Thank you for this comment. .We chose to conduct our literature review focussing on studies in Europe, in Portugal very few studies have been carried out In the European and in particular in the Portuguese context, immigrant children (1st and 2nd generations) are frequently in families with lower socioeconomic status . Health outcomes are diverse depending on context, but the general trend is that immigrant children are often at a disadvantage . We have clarified this in pg.4 lines 2-15.

Page 4 Line 40-48: I agree with the necessity of conducting research at a more regional level in a single country. But more justification is needed for the necessity of CRIAS. Currently, most birth cohort studies collected children's and parents' migration history data. So what is the strength of the CRIAS cohort that other birth cohorts cannot achieve? I think this is the most important part of the justification/rationale for this study. Maybe this reference is helpful (Grosser A, Razum O, Vrijkotte TG, Hinz IM, Spallek J. Inclusion of migrants and ethnic minorities in European birth cohort studies—a scoping review. *The European Journal of Public Health*. 2016 Dec 1;26(6):984--91.)

Thank you for raising this question and for the very relevant suggested reference. As we have replied above in the reviewers major comment on this issue , the justification for the study is now better addressed in the revised paper in Introduction pg.5 lines 13-20 and strength and limitations section pg.16-17 lines 24-4

Page 5 Line 10: pleas spell out CRIAS for the first time to use

It is in line 2 of abstract

Page 5 Line 38: please spell out AJPAS for the first time to use

page 5 line 23 Associação de Intervenção Comunitária, Desenvolvimento Social e de Saúde

Page 5 Line 42--56: I think this paragraph belongs to the Methods section about the participants. When I finish reading the setting section, I do not think “Children born in 2015, residing in the Metropolitan Lisbon Region --Amador, are followed up ...” is an accurate description, particularly for the word “followed up”, which means “children born in 2015 was recruited and baseline data were collected as well.” The authors mentioned later for the inclusion criteria “children had to be born in 2015” but the true recruitment time was 2019--2020.

This is now in the methods section pg. 6 lines 10-15.

Page 5 Line 51--52: “Measurement were scheduled at the ages of 4,5,6/7 ...” should be more accurate. If a child was born in early 2015, and the recruitment occurred in Feb 2020, then the child age is 5 at baseline. So better to use age range for each measurement timing, such as 4/5, 5/6, 6/7 ...

We fully agree with this suggestion and changed throughout the text

Cohort description

Page 6 Line 19-20: The authors mentioned the Amadora municipality is served by 10 primary health care centres. Do these 10 centres all public ones? Are there any private health care centres? Readers are not familiar with the health care system in Portugal. It would be great to briefly introduce whether it is universal health care or not.

Thank you for pointing this out. These are all public health centres. Access to private health care is limited to those with a private health insurance or health sub-system associated to certain occupations. A note has been introduced about the health system in Portugal (page 6 lines 3-8) :“The National Health Service (SNS), based on the Beveridge model, is universal and free for children up to the age of 18. Hence, healthcare arrangements in the SNS are the same for all children regardless of their migration status. They include preventive measures such as vaccination and child health monitoring assessments carried out in health centres , as well as specialist and hospital care.”

Page 6 Line 31: Did the recruitment discontinue in March 2020? If so, please make it clearer.

Yes it was discontinued – pg.6 line 12

Page 6 Line 34-42: What is the reason of selecting children who had to be born in 2015? The authors should justify why focusing on children aged 4-5 years at baseline.

The main reasons were twofold; first we wanted to have in our study children born outside Portugal (exposure is migration) and if we would have recruited at an earlier age the number of these children would be very small; second by enrolling children 2 years before school start (age 6) we were expecting to have a window of opportunity for early detection and intervention on potential identified vulnerabilities in all children of the study. This is now clearly stated on text pg. 6 lines 13-15

Also, the inclusion criteria based on the last two years' records of attending health care is another concern. That means, the recruiting samples are those who had used health services before. These children are not representative at all. We do recognise that there is a selection bias and have highlighted this as a limitation . Our decision to recruit and follow-up these children in partnership with primary care centres is based on the fact that this care is closer to the community . Most families who have a private health insurance still attend public primary care for vaccinations and administrative reasons . Visits to private sector are mostly related with specialist care. We acknowledge that this limits the representativeness to the population attending primary care

Later on, the authors mentioned using 1:1 pair method for immigrant and non-immigrant children. How did the authors pair this for non-immigrant children? Based on a random selection from the same health care centre? Or from the same type of health service use? Or based on gender?

The sample of non-immigrants is selected using a systematic sampling method in each health centre for each week; we recruit one immigrant and the non-immigrant that was before or after him in the consultation list if there is one. However, we do not have always 1:1 matching in all health centres. Because in some health centres there are more immigrant children than non-immigrant; in others is

the contrary; and in others it is 1:1. What we have is a number of children in each health centre that is proportional to the dimension of the health centre.

Instruments and measures

Page 9 Line 9-10: Please add response options in brackets for sex, age, country of birth and other variables, such as sex (male/female), age (continuous), ...

The section has been considerably reduced. Variables collected are detailed on figure 2 and some categories are shown on results tables.

Page 10 line 58-59: It would be better to describe the other group as “at risk” rather than “to monitor”. In general, when the child does not perform items in the scale, a monitoring review is suggested, usually in 6 months’ time . We would prefer to describe the group as “monitoring required” instead of labelling as” at risk”.

Page 11 Line 3-17: I agree with the collection of these indicators about the access and utilisation of healthcare services. Will the authors also consider collecting data from the patients’ perspective, such as health literacy, quality of health services? These factors are also very important to child health.

We agree with your suggestion. On future plans section we address this question in part with the plan for a qualitative study regarding the experience of parents with the health services .

Data linkage

Please spell out the abbreviations for the first time to use, including SCLINICO, SORIAN, SNS, ID . We regret but we couldn’t find info for the spelling for Sclinico and Sorian .

Table 1

It looks like non-immigrant children are more likely to come from high socioeconomic status families (university degree: 27.9% versus 20.0%; high skilled: 34.3% vs 16.0%; >\$2000: 17.8% vs 4.4%). Is the difference true between immigrant children and non-immigrant children? Or due to the selection bias (which I mentioned before for the 1:1 pair issue)? This should be discussed later.

This difference is true between immigrant and non-immigrant families, in general in Portugal. For example studies from the Migrant Observatory in Portugal show that immigrant from outside the EU (third country nationals) are in a socioeconomic disadvantage situation when compared with non-immigrants (OM, Imigração em Numeros, 2021, pag. 158). Other studies in Europe report the same – pg. 4 lines 2-5

Table 2

What is the definition of “health coverage beyond national health service”? This term was not mentioned in the Methods section.

Thank you for this question . Some families have a voluntary health insurance or an insurance scheme associated with certain occupations (eg. police force) which covers them to access private health care . This was replaced by private health insurance .

Future plans

Page 20 line 44-45: The context of “free health care for all children” should be introduced earlier before.

We agree and introduced a brief description of the health system in Portugal in the setting section (pg 6 lines 3-8)

Figure 1

Please check the accuracy of the first line entitled “children born in 2015 attending Amadora health care centres.” Did the targeted children attend Amadora healthcare centres in 2015? If so, the baseline data collection time should be 2015.

The targeted children were born in 2015, we did not collect information if they attended the health centre in 2015. To be included in the study children were born in 2015 and had registered attendance in the health centre in the 2 previous years. We clarify this question in pg. 6 lines 15-16.

Supplementary table 1

What is the p-value for the psychomotor development? “normal” versus “at risk” would be better than “all items achieved” and “to monitor” as response options of psychomotor development.

The p-value 0.155 refers to the *Pearson- Chi square statistical test with a significance level of 5%, this information was added as footnote

REVIEWER 2

Before providing a point-by-point response to the comments, the authors would like to thank the reviewer for the time and consideration in reading and commenting this paper.

The aim of the study as outlined on page 6 line 7 suggested a correlation between the exposures (which were not defined and measured) and the health outcomes of children. however, the result was mainly descriptive of the first wave of the cohort study.

Thank you for raising this point. The aim of the cohort study is to explore whether children having been exposed to a migratory process, present with different physical health risks over time and/or cognitive, socio-emotional and behavioural development challenges when compared to children born in Portugal and raised by parents also born in Portugal. This cohort profile paper focusses on the cohort description and baseline characteristics of participants. This has now been highlighted at the end of the introduction section pg.5 lines 23-24. Results from the cross-sectional study conducted during the recruitment phase to assess children’s socio-emotional and behavioural status at baseline have already been published and are briefly mentioned in the findings pg.12 lines 13-17. The next

round of follow-up data is expected to provide finer analyses of possible differences between children from immigrant families and those from non-immigrant background.

The discussion section needs more elaboration on the importance of the findings in relation to the children's health and as compared to other studies and research articles.

Thank you for your suggestion. We have highlighted the strengths of this cohort comparing with other cohorts in Europe in the strengths and limitations section. CRIAS is the first cohort study in the Metropolitan Area of the capital city Lisbon, Portugal, created to specifically address the role of immigration as a physical and emotional health determinant, by comparing outcomes in immigrant and non-immigrant children over time. In contrast to many cohort studies initiated in the EU which include on average 10% of children with a migration background, this cohort study includes about 50%. This proportion of immigrant children in the sample increases the power of comparisons between immigrant and non-immigrant children over time compared to other studies. The immigrant populations in the cohort are mainly from Brazil and Portuguese speaking African countries who might have diverse migration experiences, biological and cultural factors and health utilization behaviours than the populations included in other cohort studies in Europe.

However, we have not included a discussion section for reflection on baseline findings and comparison with other studies as it was our understanding that this is not required for this type of cohort profile paper.

Abstract

Purpose: CRIAS is a prospective cohort study created to better understand the health

trajectories of immigrant and native children in the Lisbon area, Portugal. It aims to analyse

child health determinants, focussing on migration, and identify factors associated with

physical, cognitive and social-emotional development outcomes and utilization of health services. It will be helpful to quantify the exposures and control for in the data analysis. For example, length of stay, years spent in the new country.. etc

We agree with your suggestion. Length of stay in the country for parents of immigrant children and for 1st generation immigrant children were included in the information collected and is presented on baseline results. In future follow-ups it will be included and controlled for on finer data analysis.

Participants: The original CRIAS was set up to include 604 children born in 2015, of which 50% immigrant, and their parents. We recruited 420 children between June 2019 and March 2020. Data was collected at age 4 and 5 years; follow-up at age 6/7 is under way. Then this report is only a cross sectional study not a prospective cohort.

This report describes the characteristics of the cohort CRIAS, the baseline cross-sectional results and the 1st follow-up study's main findings . This has been clarified pg.5 lines 23-24.

Introduction

A rapidly growing part of the population in Europe is composed of immigrants. In 2020, Portugal registered its highest number: 6.4% of the country's population were immigrants mostly non-EU nationals (69%); Brazil (28%) and Portuguese speaking countries in Africa are the main countries of origin, but an increasingly significant number arrived from Asia [1]. Consider rewriting this sentence.. keep it clear and simple.

The sentence has been rewritten pg.3 lines 7-10.

However, results are not always consistent – this might reflect the heterogeneity of the immigrant groups between countries or even among regions. Hence the importance of conducting research at a more regional level within a single country.. Not sure what this sentence indicate?? if it is the gap, then it should be highlighted more.

Thank you for the opportunity to clarify this point. That sentence has been removed and we further developed the importance of the diversity of origins and contexts of immigrant populations . There is a diversity of origins and contexts in immigrant populations in Europe and therefore studies results on health outcomes in immigrant children are not always concurring (pg.4 line 6-7). For example, due to its colonial past, in Portugal immigrant populations are mainly from Brazil and Portuguese speaking African countries who might have diverse migration experiences, biological and cultural factors and health utilization behaviours than the populations included in other cohort studies in Europe.

Childhood, especially the first 8 years, encompasses a rapid period of growth and development which plays a key role for health and wellbeing across the life course [19]. This period is highly influenced by the environment where the child grows and develops, namely by socioeconomic factors (20) This is

very true, I would to encourage the authors to explore the catch up growth and windows of opportunities body of literature and reflect how it might related to their current work.

Thank you for your suggestion. We have introduced a paragraph on opportunities for interventions at this age group which included nutrition pg. 4 lines 26-29.

The possibility to formulate and implement early interventions. These can help children not only to reach their full potential when starting school, but also to have a positive impact on their future health I think authors can be more specific here and add a few references and research articles where early interventions in childhood can overcome and to some extent revers the impact of early childhood experiences. It seems the authors are focusing on the physical health outcomes (growth and development, obesity... etc) if so a literature review that address these health outcome is necessary.

We agree with your suggestion and gave reference to examples in the UK (page 4 line 30-31). We take the opportunity here to mention that we are focussing on the role of migration as much on physical as on emotional/behavioural outcomes over time .

The aim of the CRIAS cohort is to explore the effects of exposures, focussing on

migration, and identify risk factors associated with the physical, cognitive and social emotional development outcomes, as well as with access and utilization of healthcare services. The aim should be clear and simple, define the exposure, migration, physical, cognitive, and emotional development.

Also exposures are very vague term are they concerned with environmental exposures...etc We have rewritten the aim of the CRIAS cohort study in pg.5 lines 13-20: “ The CRIAS cohort is the first longitudinal study in the Metropolitan Area of the capital city Lisbon that specifically focuses on gaining a better understanding of the health and development trajectories of immigrant and non-immigrant children, given their respective socioeconomic and cultural contexts. The aim of the CRIAS cohort is to explore whether children exposed to a migratory process, present with different physical health outcomes, cognitive, socioemotional and behavioural challenges and with different health care utilization patterns, over time, when compared to children born in Portugal and raised by parents also born in Portugal.”

Cohort description

There were 1009 children with these characteristics. please clarify what are these characteristics.

These characteristics relate to being born in 2015 and have recorded attendance at the health centre in the last 2 years, they correspond to the eligibility criteria. We replaced “characteristics “ with “eligibility criteria.”

Based on a previous study [24] we assumed that around 30% of users are immigrant children it will be interesting to understand how they made this assumption.

In the study referenced, the sample obtained suggested that among the children users of the health centres, 30% are immigrant .

Instruments and measures This section can be shortened to a few sentences only.

Thank you for your suggestion. This section has been significantly reduced, however information that we considered relevant regarding some of the instruments was kept on main text.

Results

From the cross-sectional data collected at baseline, low parental education level (aOR 2.5; 95%CI: 1.11- 5.16) and being a 1st generation immigrant child (aOR 2.2; 95%CI: 1.06-4.76) appeared to increase the odds of developing emotional and behavioural difficulties [35]. I am not sure how they reached this conclusion. I dont see any OR in the tables and why the references 35 is used here.. Please clarify and provide the relative table if necessary.

As per your suggestion we have included a table with regression results as supplementary table 1.b. These results from the cross-sectional data at baseline have been published and the reference given (now ref. 45) corresponds to the article and is given on pg. 14 line 2.

Overweight was found in 25% of the children (22% in immigrant vs 28% in native children), 6% of children were obese and from a total of 8% underweight children, 72% were immigrant it will be interesting to report if the obesity and underweight coexist in the same families or not and how the obesity correlated with the lenght of stay in the new country.

Up to now we did not collect anthropometric data on the families.

I didnt see anything about the expsoures and how it correlated with the health outcomes. The aim of the study indicate there will be correlation calculation and prediction.

We hope to have clarified this point previously by rewriting the aim of the CRIAS cohort and by specifying the aim of this paper pg.5 lines 23-24: “This paper describes the characteristics of the cohort, the baseline cross-sectional study’s and the 1st follow-up main findings.”